# The Quality of Carer–Patient Relationship Scale: Adaptation and Validation into Portuguese

**DOI:** 10.3390/ijerph18031264

**Published:** 2021-01-31

**Authors:** Rosa Silva, Elzbieta Bobrowicz-Campos, Paulo Santos-Costa, Isabel Gil, Hugo Neves, João Apóstolo

**Affiliations:** 1Health Sciences Research Unit: Nursing of the Nursing School of Coimbra, 3004-011 Coimbra, Portugal; elzbieta.campos@gmail.com (E.B.-C.); paulocosta@esenfc.pt (P.S.-C.); igil@esenfc.pt (I.G.); hugoneves@gmail.com (H.N.); apostolo@esenfc.pt (J.A.); 2Centre of 20th Century Interdisciplinary Studies, Faculty of Psychology and Educational Sciences, University of Coimbra, 3000-115 Coimbra, Portugal

**Keywords:** aged, carers, dementia, quality of caregiver-patient relationship scale, reliability and validity

## Abstract

Background: This study aimed to translate and adapt the Quality of the Carer–Patient Relationship (QCPR) scale into Portuguese and analyse both its psychometric properties and correlation with sociodemographic and clinical variables. Methods: Phase (1) Translate and culturally adapt the scale. Phase (2) Assess the scale’s confirmatory factorial analysis, internal consistency, construct validity, and correlations. Results: The experts classified the overall quality of the translation as adequate. A total of 53 dyads (cared-for person and carer) were assessed. In both versions, measures of central tendency and symmetry were also adequate, and the two factors under investigation had appropriate reliability, although in the conflict/critical factor, this was more fragile. Cronbach’s alpha values were 0.89 for the cared-for person version and 0.91 for the carer version. Conclusions: The QCPR scale showed satisfactory to good values of reliability. The assessment is essential to guarantee structured interventions by health professionals, since the quality of the dyads’ relationship seems to influence both older adults’ quality of life and carers’ health status. This study is a significant contribution to the introduction of the QCPR scale in the Portuguese clinical and scientific culture but also an opportunity to increase its use internationally.

## 1. Introduction

Ageing is intrinsically associated with the gradual decline of functional capacity, whether physical or cognitive, resulting in the need to establish support networks for older adults living at home [1].

Being a carer of someone with a neurocognitive disorder (NCD), commonly referred to as dementia [2], can be a strain in many dimensions, including emotional, social, and physical [3,4]. In addition, there are often various neuropsychiatric symptoms (NPS) that accompany this degenerative process such as restlessness, mood disturbances, and disorientation, which increase the complexity of informal care [4,5]. The presence of such NPS can lead to significantly more physical and emotional demand on the carer, which can affect the quality of the carer’s relationship with the cared-for person [6,7,8]. The fragmentation of the relationship between the dyad (cared-for person, and carer) will have implications for the quality of care itself, as well as for triggering negative feelings that (when not identified and accompanied by professionals) can result in depressive symptoms or other mental illness in the carer and/or person with NCD and a lower quality of life [9]. A lower quality care relationship also increases the chance of early or even unnecessary institutionalisation [10].

In the international literature, there is a scarcity of instruments that focus on assessing the quality of the relationship between the cared-for person with NCD and carer [11,12]. Spruytte and collaborators [7] proposed an instrument—the Quality of the Carer–Patient Relationship (QCPR) scale—to assess the relationship quality between the members of the dyad. The QCPR scale, originally a Belgian instrument, written in the Dutch language [7,12], has been translated and validated by its authors into the English language [12]. Since 2000, the QCPR scale has been used in several studies carried out by its authors. First, it was used to evaluate the expression of emotions in families of patients with schizophrenia, and then it was adapted for families of patients with dementia [8,10,13,14]. Recently, the QCPR scale has been used in studies in the area of cognitive stimulation, being administered by a formal caregiver at home with the purpose of monitoring the quality of the relationship between members of the dyad [15,16,17].

With the intent of assessing the relationship’s quality through the perceptions of the elements of the dyad, the QCPR scale includes two versions, one for the cared-for person and another for the carer. Each version assesses two dimensions of the emotion expressed: warmth/affection, the positive dimension; and conflict/criticism, the negative dimension [7,12].

Given this scarcity of scales to assess the emotion expressed within the dyad, research in this field is considered to be essential. Therefore, it is vital to ensure that measurement tools for clinical practice in nursing are available, as these facilitate the assessment of the quality of the relationship within the dyad. As such, the identification of the relationship profile will enable the best care decisions to be implemented. This study sought (i) to translate and adapt the QCPR scale into European Portuguese and its associated culture, as well as analyse its psychometric properties and (ii) to describe the sociodemographic (gender, age, cohabitation) and clinical variables of interest of the study sample (cognitive performance, NPS, and quality of life) and analyse their correlation with the new version of the QCPR scale.

## 2. Materials and Methods

The study was developed in two phases: Phase 1, which included the translation and cultural adaptation of the QCPR scale into European Portuguese (QCPR-P); and Phase 2, which encompassed the validation of the psychometric properties of the new version of the scale (QCPR-P).

### 2.1. Phase 1: Cultural Adaptation of the QCPR

The process of culturally adapting the QCPR followed the international guidelines for the translation and adaptation of self-report measures [18]. Initially, the English version of the QCPR was translated into Portuguese by two independent Portuguese translators fluent in English (forward translation). Then, a panel composed of two researchers (both nurses, one PhD student, and another MSc) compared the two translated versions with the original and, based on the adequacy and comprehensibility of the content, developed a first preliminary European Portuguese version of the QCPR (consensus version α). This preliminary version was sent to two independent translators, who were native in English and fluent in Portuguese, who translated it back into English (back-translation). Both back-translations were analysed by the research team, including Nele Spruytte. The combination of items with the best back-translation, resulting in consensus version β, was then subjected to an analysis of content validity concerning its semantic, idiomatic, conceptual, and cultural dimensions.

An expert review committee composed of nurses (n = 8) came together to analyse the content validity of QCPR version β. All committee members (n = 8) have expertise in the field of ageing, cognitive impairment, and the care of cognitively declined persons alongside the involvement of a family support system. All committee members (n = 8) were also proficient users of the English language. 

The analysis of the content validity was performed independently by each committee member. This analysis considered several parameters, including textual clarity, appropriateness, relevance, and linguistic equivalence, for the different QCPR sections (introduction, instructions for scale completion, items, and scoring system). Within each section, the parameters were classified using a seven-point Likert scale with responses ranging from totally disagree (1) to totally agree (7). In addition, the committee experts were given an open question about the overall quality of the QCPR version β (in which they had the opportunity to provide suggestions to reformulate any QCPR parameter they deemed relevant). The committee experts were also questioned regarding whether they would recommend the QCPR version β for use in clinical practice. 

For each QCPR parameter, the index of content validity (ICV) was calculated based on the formula: ICV = number of responses “agree” and “totally agree” /number of all responses [19]. Only parameters with an ICV ≥ 0.8 were considered to be sufficiently satisfactory to be included in the final version of the QCPR [19]. The QCPR parameters with an ICV < 0.8 were reformulated according to the committee experts’ suggestions and subjected to a new analysis [19]. After reaching the required ICV threshold for all QCPR parameters, the analysis of the content validity was concluded, and the final version of the QCPR-P was achieved.

### 2.2. Phase 2: Validation of the European Portuguese Version of the QCPR

#### 2.2.1. Sample

The psychometric properties of the culturally adapted version of the QCPR-P were tested in a convenience sample recruited from two primary healthcare units. The sample included 53 dyads, each constituted by a person with NCD and his/her carer (n = 106). To be considered eligible to participate in the study, the cared-for person had to be aged 60 years or more, live in his/her own house, and be diagnosed with mild or major NCD by a neurologist or psychiatrist. To ensure that the sample of the cared-for persons did not include adults with severe cognitive decline, the 6-item Cognitive Impairment Test (6-CIT; [20]) was additionally administrated. The 6-CIT is a cognitive screening test, and all participants who scored ≥21 points on the 6-CIT were excluded from the study. As for the carers, to be included in the study, they should have the capacity to communicate, and the presence of cognitive impairment was considered a reason for exclusion.

#### 2.2.2. Measurement Scales

The QCPR scale displayed 14 items, some of which are symmetrical (e.g., “The person and I…”), while others are asymmetrical (e.g., “I …”). An important advantage of the scale is its applicability for several kinship relationships such as children, partners, etc. During its validation process, the original QCPR scale displayed appropriate psychometric properties and good internal consistency, with a Cronbach’s alpha of 0.82 [7,12]. All 14 items are classified on a 5-point Likert scale ranging from “totally disagree” to “totally agree”. For the warmth/affection items, the “totally disagree” responses were scored with one point and the “totally agree” responses were scored with five points. The intermediate responses were scored with two, three, or four points. The items from the conflict/criticism subscale were classified inversely. In the original QCPR, a score under 42 reflects a poor relationship, and a score over 56 reflects a good relationship. The remaining scores (between 42 and 56) indicate that the quality of the relationship is common, i.e., a standard relationship [7]. Whenever necessary, the evaluator can assist the participants in reading the scale items.

Both the cared-for person and his/her carer completed the QCPR-P separately. Another instrument administered to both the cared-for person with NCD and his/her caregiver was the Quality of Life in Alzheimer’s Disease scale (QoL-AD; [21]). The battery of tests administered to the cared-for persons with NCD also included the following: (i) the Alzheimer’s Disease Assessment Scale-Cognitive subscale (ADAS-Cog; [22]), which evaluates the severity of cognitive dysfunctions; (ii) the Neuropsychiatric Inventory (NPI; [23]), which is designed to assess the frequency and severity of behavioural disturbances that occur in people with NCD. Additionally, carers were given the Short Form-12 Health Survey (SF-12; [24]). This is a self-reported measure designed to evaluate the status of health in its physical and mental dimensions.

#### 2.2.3. Procedures

The present study was carried out as part of a larger project aiming to evaluate the effectiveness of the cognitive stimulation programme for older adults with NCD implemented by their carers. Subsequently, all dyads included in the sample were followed in the intervention process. The first contact with dyads was established by health professionals (medical doctors and nurses) working in primary healthcare units. At that time, the dyads were provided with a brief presentation of the study objectives and were asked permission to be contacted by a research team. A research team member, who presented the specific goals of the study and scheduled a face-to-face meeting if the dyads maintained their interest in participating, contacted all the dyads that showed interest in participating in the study by telephone. During this conversation, informed consent from each dyad member was obtained, and the eligibility criteria for participation in the study were verified. The assessment process occurred in participants’ homes and data were collected from April to September 2017 by two health professionals trained in psychogeriatric assessment.

#### 2.2.4. Ethical Considerations

All procedures performed in studies involving human participants were in accordance with the ethical standards of the institutional and/or national research committee and with the 1964 Helsinki declaration and its later amendments or comparable ethical standards. All subjects who agreed to participate in the study gave their written and informed consent. The study was approved by the Ethics Committee of the Administração Regional de Saúde do Norte (Protocol number 27/2017).

#### 2.2.5. Data Analysis

The internal consistency of the QCPR-P was assessed using the Cronbach’s alpha coefficient and the corrected item-total correlation. A Confirmatory Factorial Analysis was conducted to assess the overall adjustment quality of the factor model. To improve the adjustment quality of the model, analysis of the modification indices was conducted, with the establishment of correlation between the identified errors. Convergent validity was calculated by the mean extracted variance (MEV ≥ 0.5); discriminant validity was also calculated by the MEV factor (i and j). By combining these three components (factorial, convergent, and discriminant validity), the construct validity of the instrument was evaluated [25].

As the outcome variables did not follow a normal distribution, data analysis was performed using non-parametric methods. The associations between continuous variables were examined through Spearman correlations. Comparisons involving two subgroups were performed based on the Mann–Whitney test; to analyse differences between more than two subgroups, the Kruskal–Wallis test was used. The effect size was estimated according to the following formula: r = Z / √N [26]. Probability levels of 0.05 were considered significant. Data were analysed using the SPSS Statistics v. 24 program, including AMOS software (IBM SPSS, New York).

## 3. Results

### 3.1. Phase 1: Cultural Adaptation of the QCPR

The first analysis of the QCPR version β resulted in the identification of several aspects that needed to be improved. The committee of experts suggested the reformulation of items 1, 3, 11, and 14, and changing the name of the scale; for example, the term “patient” was replaced with “cared-for person”. They also proposed rephrasing instructions for completing and scoring the scale. After the correction of QCPR version β according to the suggestions of the expert review committee, the cultural adaptation of the QCPR was successfully completed. Twelve of the 14 items included in the final version of the QCPR achieved ICV > 0.80. The remaining two items (items 1 and 14) obtained ICV = 0.80. All committee experts classified the overall quality of the QCPR translation as wholly adequate. The 100% agreement was also obtained in relation to the recommendation of the scale for use in clinical practice.

### 3.2. Phase 2: Validation of the European Portuguese Version of the QCPR (QCPR-P)

The sociodemographic and clinical characteristics of the sample are presented in Table 1.

#### 3.2.1. QCPR-P Comprehensibility

During the administration of the QCPR-P, some of the cared-for persons with NCD had difficulty understanding the first two items in the scale, and in choosing between the following response categories: “totally agree”/“agree” and “disagree”/”totally disagree”. The interviewer responded to these difficulties by reading the questionnaire slowly and explaining all the parts that were considered less clear. The carers did not report any difficulties in understanding and completing the QCPR-P.

#### 3.2.2. QCPR-P Score

From the perspective of the persons with NCD, the average relationship quality with the carers was shown to be tangential between common and good [M = 55.66 (±7.16)]. In relation to the carers, their relationship with the cared-for persons was classified on average as being of common quality [M = 54.26 (±9.25)]. For more details, see Table 2.

Only 62% of the classifications made by persons with NCD and their carers were shown to be congruent. In 23% of the cases, the assessment of the quality of the dyad’s relationship made by persons with NCD was more positive than the assessment performed by the caregivers. In the remaining 15% of cases, the rating made by the carers was more positive than the rating made by the cared-for person.

#### 3.2.3. QCPR-P Internal Consistency

The internal consistency of both versions of the QCPR-P scale was high. The Cronbach’s alpha estimated for the QCPR-P version for the cared-for persons was 0.886, with item-total correlations ranging from 0.36 (item 13) to 0.80 (item 4). Only one of the item-total correlations was shown to be weak (r < 0.40; item 13), four were moderate (0.40 ≤ r < 0.70; items 3, 10, 11, and 14), and the remaining nine (items 1, 2, 4–9, 12) were strong (r ≥ 0.70). Regarding the QCPR-P version for carers, the Cronbach’s alpha obtained a value of 0.911. The item-total correlations were either moderate (items 1, 4, 5, 10–12) or strong (items 2, 3, 6–9, 13, and 14), and varied from 0.41 (item 10) to 0.74 (item 7). Other item-total statistics are presented in Table 3.

#### 3.2.4. Overall Quality of Model Fit-QCPR-P

The quality of the model fit for the cared-for person version is good for all index values, with the exception of the Parsimony Goodness-of-Fit Index (PGFI) and Parsimony Comparative Fit Index (PCFI), which show a poor to reasonable fit. Since the X2 index has a *p* > 0.05, it can be assumed that the covariance matrix of the tested model does not deviate from the theoretical model. In turn, the Root Mean Square Error of Approximation (RMSEA) values determine a good fit (Method for Eclipsing Component Identification before adjustment = 4.097 vs. after adjustment = 3.414), which allows us to measure the stability of the adjusted model (see Figure 1)

Regarding the quality of the model adjustment for the carer version, except for the Goodness-of-Fit Index (GFI) and its parsimony index, which suggest a poor fit, the remaining parameters show good model fit. The X2 index allows us to assume that the covariance matrix of the tested model did not deviate from the theoretical model (*p* > 0.05). The Modified Expected Cross-Validation Index (MECVI) value was higher before the adjustment (4.097 vs. 3.414), which enabled us to measure the stability of the adjusted model (see Figure 2).

#### 3.2.5. Composite Reliability

In the QCPR-P version for the cared-for person, the two factors under investigation (warmth/affection and conflict/criticism) have appropriate composite reliability (CR). The warmth/affection factor achieved good CR (=0.90), although the conflict/critical factor obtained lower reliability, albeit above the acceptable limit (=0.72).

In the QCPR-P (carer version), the two factors under investigation (warmth/affection and conflict/criticism) had appropriate CR. The warmth/affection factor shows good CR (=0.93), although the conflict/criticism factor has lower reliability, albeit above the acceptable limit (=0.76).

#### 3.2.6. Construct Validity

All QCPR-P (cared-for person version) items had factor loadings within the acceptable range (λ ≥ 0.50), except for item 13 (λ = 0.22). Individual reliability was also adequate, again except for item 13 (R2 ≥ 0.25). MEV showed adequate convergent validity for the warmth/affection factor (MEV = 0.54) but not for the conflict/criticism factor (MEV = 0.31). The discriminant validity was not demonstrated, given that the estimated MEV (0.98) of the factors was below the square of the correlation between these factors (R2 = 0.95). Thus, construct validity (which depends on factorial, convergent, and discriminant validity) was not demonstrated. In conclusion, the operationalisation of the latent construct was not revealed.

All QCPR-P (cared-for person version) items have factor loadings within the acceptable range (λ ≥ 0.50), except for items 10 and 11 (λ = 0.38 and 0.33, respectively). Individual reliability is also adequate for all items (R2 ≥ 0.25), but again except for items 10 and 11 (R2 = 0.14 and 0.11, respectively). The MEV revealed adequate convergent validity for the warmth/affection factor (MEV = 0.62), but the same was not verified for the conflict/criticism factor (MEV = 0.36). The QCPR-P (carer version) showed discriminant validity, since the squared MEV (MEV= 0.67/0.68) of the factors (warmth/affection and conflict/criticism) is higher (or equal) than the correlation of these factors (R2 = 0.67). Due to the factorial and convergent validity of the conflict/criticism factor, the validity of the latent construct was not demonstrated.

#### 3.2.7. Distribution of the QCPR-P Score in the Different Subgroups

Gender: Regarding the QCPR-P (cared-for person with NCD version), significant differences were found in the conflict/criticism subscale score (U = 173.00, *p* < 0.05, *d* = 0.638), with female participants obtaining higher mean ranks (n = 38, mean rank = 29.95) than male participants (n = 15, mean rank = 19.53). There were no significant differences between the total scores of the two subgroups (U = 200.50, *p* > 0.05, *d* = 0.471) nor in their warmth/affection subscale scores (U = 251.50, *p* > 0.05, *d* = 0.182).

Regarding the QCPR-P (carer’s version), significant differences were found between male and female participants in the distribution of their warmth/affection subscale scores (U = 140.50, *p* < 0.05, *d* = 0.567), with mean ranks being higher for women (n = 42, mean rank = 27.54) than for men (n = 11, mean rank = 18.77). However, the distribution of the total score and conflict/criticism subscale score did not differ in the subgroups of male and female carers (total score: U = 168.00, *p* > 0.05, *d* = 0.387; conflict/criticism subscale: U = 208.50; *p* > 0.05, *d* = 0.136).

Cohabitation: From the perspective of the person with NCD, the distribution of the QCPR-P total scores in both subgroups did not differ significantly (U = 239.50, *p* > 0.05, *d* = 0.435). The lack of significant differences between persons with NCD who cohabit and who do not cohabit with their carers was also verified for the QCPR-P warmth/affection and conflict/criticism subscales (warmth/affection subscale: U = 240.00, *p* > 0.05; *d* = 0.433; conflict/criticism subscale: U = 243.00, *p* > 0.05, *d* = 0.416).

The analyses performed for the QCPR-P (carer’s version) identified that there are no significant differences in the distribution of the total score and the warmth/affection subscale score in the subgroups of carers who cohabit and who do not cohabit with the cared-for persons (total score: U = 238.00, *p* > 0.05, *d* = 0.444; warmth/affection subscale: U = 267.00, *p* > 0.05, *d* = 0.288). However, significant differences were found in relation to the conflict/criticism subscale score (U = 267.50, *p* < 0.05, *d* = 0.286). The carers living with cared-for persons had a lower mean rank (=23.28) than the carers who lived separately (=32.58).

#### 3.2.8. Correlations between QCPR Score and Age

The age of the person with NCD was not significantly correlated with the total score of the QCPR-P (cared-for persons version; r = 0.022). The absence of significant associations was also confirmed for the ages and scores obtained in both subscales of the QCPR-P (cared-for person version; warmth/affection subscale: r =−0.028; conflict/criticism subscale: r = 0.048). As for carers, the total and subscales scores of the QCPR-P (carer’s version) were significantly and negatively correlated with age. In all cases, the magnitude of the correlation was weak (total score: r = −0.332, *p* < 0.05; warmth/affection subscale: r = −0.290, *p* < 0.05; conflict/criticism subscale: r = −0.353, *p* < 0.01).

#### 3.2.9. Correlations between the QCPR-P and Variables of Individual Functioning

The outcomes of interest related to the functioning of the person with NCD considered in this section were cognitive performance evaluated based on the 6-CIT, ADAS-Cog, NPS (measured through the NPI), as well as quality of life appraised using the QoL-AD. The descriptive statistics relative to these outcomes of interest are provided in Table 4.

As can be seen in Table 5, the scores obtained by the cared-for person with NCD in the 6-CIT, ADAS-Cog, and NPI were not significantly correlated with the score (total and for the two subscales) obtained in the QCPR-P (cared-for person with NCD version). However, the total score of the QCPR-P (cared-for persons version) was significantly correlated with the QoL-AD results. The correlation identified was positive and of weak magnitude. There were also significant positive correlations between the QoL-AD and the warmth/affection subscale of the QCPR (cared-for person with NCD version) and between the QoL-AD and conflict/criticism subscale of the QCPR (cared-for person with NCD version). The first one was of moderate magnitude and the second was of weak magnitude.

Regarding the carers, the study of correlations considered their perception of the quality of their relationship with the person with NCD and the health status evaluated through the SF-12. The significant positive and weak correlations were found between the total score of the QCPR–P scale (carer’s version) and the physical and mental dimensions of SF-12. The conflict/criticism subscale of the QCPR-P scale (carer’s version) also significantly correlated with the physical and mental dimensions of SF-12. However, in this case, the magnitude of association was moderate. There were no significant correlations between the QCPR-P warmth/affection subscale score and the SF-12 score.

## 4. Discussion

The feeling of mutuality in a dyadic relationship is an important aspect to be discussed. Recent studies confirm the need for instruments to assess the emotion expressed in such dyads [11]. In the qualitative phase of this study, the stages of translations, retroversion, and analysis were crucial to produce and maintain the objectivity of the QCPR translation and cultural adaptation process. Although the QCPR-P is a short instrument, items such as “My relative often annoys me” and “I blame my relative for the cause of my problems” warranted greater linguistic and cultural attention to remain equivalent to the original instrument.

The dyads involved in this study share similar sociodemographic characteristics with the participants of other international studies focussed on this thematic scope [7,8]. As an example, in this study, the majority of the carers were direct descendants of the persons with NCD, were mostly female, and lived in the same household. However, in our study, the dyad’s level of education was lower than in other international studies [7,8]. This relates to a globally low educational level in the older Portuguese population, and this sample only reflects this same characteristic [27]. These low levels of education may have affected the comprehensibility of the scale items, especially in cared-for persons with NCD.

The study discovered that persons with NCD tend to evaluate the quality of the relationship in a more positive light than their carers, which corroborates the results obtained by Vermeulen and colleagues [8,10]. Several justifications can be found for this phenomenon. One possible reason relates to the method in which the QCPR scale is administered. While the majority of the carers can complete the instrument independently, persons with NCD may require assistance from external elements and conceal their feelings due to social desirability or self-protection [10]. Another justification may be that the person with NCD may answer the QCPR from a gratitude point of view, while carers may be influenced by feelings of emotional and physical overload and saturation [6]. These findings show the importance of obtaining the perspective of both elements of the care relationship, since one-sided assessments may condition proper diagnostics and structured interventions by health professionals [28].

The carer’s version of the QCPR has good acceptability, while the acceptability of the cared-for person’s version is satisfactory. This could be because the cared-for person with NCD displayed difficulties in choosing between similar response categories (e.g., “totally agree” and “agree”). This perception is confirmed by the fewer number of scores at the far ends of the Likert scale in their version of the QPCR-P; this may be explained by the gradual compromise of the decision-making ability experienced by persons with NCD [28,29], which is further aggravated by low education levels. Consequently, the consistency and reproducibility properties of the QCPR-P scale were moderate for the persons with NCD’s version and high for the carers’ version. The QCPR-P’s validity to accurately measure the construct it proposes to assess has some weaknesses. We believe that these are due to the lack of heterogeneity in the typology of the quality of the relationship (tendency to good relationship, as already reflected) and to the sample size. It is important to note that the correlation between the two factors in QCPR-P (cared-for person version) appear to indicate the presence of a single factor model. This may be due to the sample size, indicating the need for further analysis of the instruments to understand if the results obtained will be replicated.

For both versions, the conflict/criticism subscale shows acceptable, albeit fragile, reproducibility properties, and it has weaknesses in terms of criterion validity in some items. Thus, the convergent validity of this subscale has not been demonstrated. Regarding the warmth/affection subscale, for both versions, criterion validity and internal consistency have been demonstrated. Confirmatory factor analysis facilitated the evaluation of the quality of the global fit of the factorial model, and it was concluded that the model is stable for most of the evaluated parameters and, hence, a significant contribution is made in testing the validity of the theoretical model.

In our study, the results of the QCPR-P (cared-for persons with NCD version) correlated with the perceptions of these persons on their quality of life. Considering that one of the dimensions of quality of life is relationships, these results are a good indicator of the consistency of the data [21,30]. In this study, the level of cognitive impairment and the presence of NPS did not influence the distribution of the overall QCPR-P scores, which in this context are understood as a good indicator of the instrument’s reliability. The absence of this correlation was also identified by Vermeulen and colleagues [8].

However, the overall scores for the carers’ version of the QCPR-P were influenced by their physical and mental health status. The conflict/criticism subscale was inversely related to the carer’s health status in its two dimensions (mental and physical). Therefore, carers with compromised health tend to experience worse relationships with the person with NCD, which is ruled by increased conflict and criticism [6,7,8,28] and possibly less gratification in the performance of their role [11,31].

Analogously to the findings by Spruytte and colleagues [7], the overall QCPR-P scores were inversely related to the carers’ age, with younger carers achieving a more favourable relationship with the cared-for person with NCD. In turn, female carers have been shown to establish more warmth/affection in these relationships than male carers. Cohabitation and kinship were variables that influenced the quality of the established relationship. There was less conflict/criticism for non-cohabiting carers and greater conflict/criticism in dyads in which the carers were daughters-in-law [10].

Nonetheless, some limitations are worth mentioning, such as the small sample size and its homogeneity. The sample in this study was composed of dyads who subsequently participated in a program of cognitive stimulation. This reason may have biased the sample selection, since dyads willing to enroll in such studies may have closer bonds and display more positive relationships. Therefore, in the future, it is crucial to study this instrument in more heterogeneous, random samples.

## 5. Conclusions

Overall, the QCPR-P scale has shown acceptable psychometric properties, being a useful tool for the assessment of the quality of the relationship between carers and cared-for persons with NCD who reside in community settings. Despite assessing more sensitive and private dimensions, the QCPR-P scale was well received by the study participants. However, the reliability and validity values of the conflict/criticism subscale of the QCPR-P require further validation studies.

## Figures and Tables

**Figure 1 ijerph-18-01264-f001:**
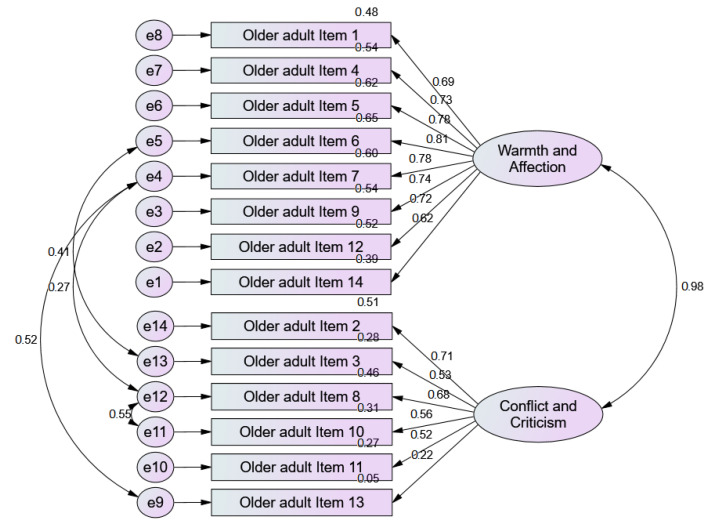
Confirmatory Factor Analysis of QCPR-P (cared-for person version).

**Figure 2 ijerph-18-01264-f002:**
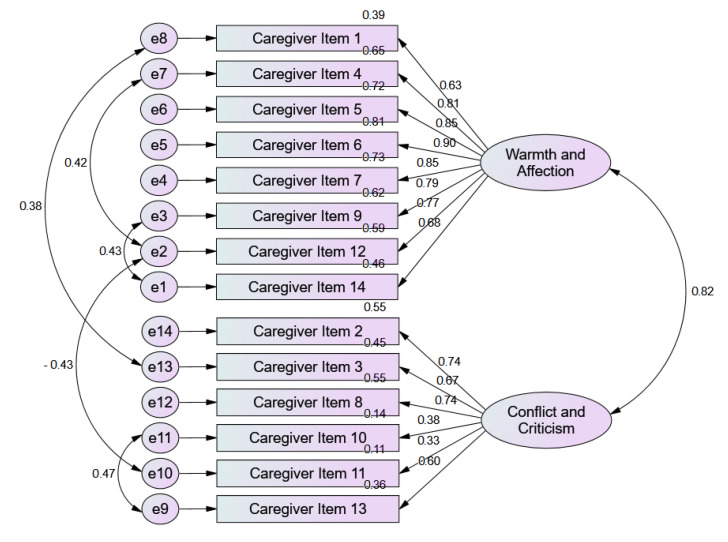
Confirmatory Factor Analysis of QCPR-P (carer version).

**Table 1 ijerph-18-01264-t001:** Sociodemographic and clinical characteristics of study participants.

Sociodemographic and Clinical Characteristics	Cared Person	Carer
Gender (male/female)	15/38	11/42
Age (Mean ± SD; range)	79.20 ± 8.90; 60–97	58.13 ± 14.61; 20–81
Education level, years (Mean ± SD; range)	4.09 ± 2.44; 0–15	7.73 ± 4.10; 3–17
Deterioration degree (%)	56.6% mild NCD 43.4% moderate NCD	- -
Kinship degree with cared person (%) Son/Daughter Spouse Grandchild Paid Caregiver Daughter in law Neighbour	- - - - - -	52.8% 26.4% 7.5% 7.5% 3.8% 1.9%
Cohabitation with cared person (%)	-	64.2%

NCD: Neurocognitive disorder; SD: Standard deviation.

**Table 2 ijerph-18-01264-t002:** Descriptive statistics of the quality of the carer–patient relationship scale.

Scale’s Scores (Total and per Subscale)	QCPR (Cared Person’s Version)	QCPR (Carer’s Version)
Warmth subscale score (Mean ± SD; range)	33.32 ± 3.82; 23–40	32.43 ± 5.70; 11–40
Conflict/criticism subscale score (Mean ± SD; range)	22.34 ± 3.79; 15–30	21.83 ± 4.55; 10–30
Total QCPR score (Mean ± SD; range)	55.66 ± 7.16; 39–70	54.26 ± 9.25; 21–70
Typology of the relationship quality (%)	5.7% poor quality 43.4% common quality 50.9% good quality	7.5% poor quality 47.2% common quality 45.3% good quality

QCPR: Quality of the Carer–Patient Relationship scale; SD: Standard deviation.

**Table 3 ijerph-18-01264-t003:** Individual items and total Quality of the Carer–Patient Relationship scale (both cared-for person and carer versions) statistics.

Item	Item Mean	Item Standard Deviation	Item-Total Correlation	Corrected Item-Total Correlation	Alpha If Item Deleted
	Cared Person’s version				
1	4.17	0.64	0.72	0.65	0.88
2	3.83	0.87	0.77	0.68	0.87
3	4.02	0.72	0.56	0.47	0.88
4	4.17	0.73	0.80	0.67	0.87
5	4. 13	0.56	0.70	0.70	0.88
6	4.19	0.59	0.72	0.77	0.87
7	4.04	0.83	0.77	0.78	0.87
8	3.81	1.06	0.75	0.69	0.87
9	4.19	0.59	0.72	0.68	0.88
10	3.42	1.12	0.64	0.55	0.88
11	3.92	1.05	0.63	0.48	0.88
12	4.17	0.47	0.70	0.69	0.88
13	3.53	1.15	0.36	0.20	0.90
14	4.26	0.49	0.60	0.54	0.88
	Carer’s version				
1	3.75	0.92	0.64	0.65	0.90
2	3.49	1.03	0.70	0.67	0.90
3	4.04	0.96	0.71	0.64	0.90
4	3.94	0.99	0.69	0.71	0.90
5	3.85	0.95	0.69	0.71	0.90
6	4.26	0.79	0.70	0.81	0.90
7	4.00	0.98	0.74	0.77	0.90
8	3.66	1.14	0.73	0.67	0.90
9	4.15	0.84	0.70	0.78	0.90
10	3.06	1.15	0.41	0.37	0.92
11	4.49	0.82	0.54	0.34	0.91
12	4.13	0.81	0.63	0.62	0.91
13	3.26	1.40	0.70	0.56	0.91
14	4.32	0.67	0.71	0.64	0.91

**Table 4 ijerph-18-01264-t004:** Descriptive statistics relative to outcomes of interest in participants with neurocognitive disorders and their carers.

Descriptive Statistics	Cared Person				Carer	
	6-CIT	ADAS-Cog	INP	QoL-AD	SF-12 Physical	SF-12 Mental
Mean (SD)	13.68 (5.65)	20.17 (7.53)	11.31 (10.73)	25.97 (5.67)	55.62 (17.13)	52.12 (16.66)
Range	4–20	4–42	0–49	15.33–38.67	20.00–82.67	9.52–80.95

6-CIT: 6-item Cognitive Impairment Test; ADAS-Cog: the Alzheimer’s Disease Assessment Scale–Cognitive subscale; INP: the Neuropsychiatric Inventory; QoL-AD: Quality of Life in Alzheimer’s Disease; SF-12 physical: the Short Form-12 Health Survey–physical dimension; SD: Standard Deviation; SF-12 mental: the Short Form-12 Health Survey–mental dimension.

**Table 5 ijerph-18-01264-t005:** Correlations between the Quality of the Carer–Patient Relationship scale and other outcomes of interest.

	Cared Person’s Version				Carer’s Version	
QCPR	6-CIT	ADAS-Cog	INP	QoL-AD	SF-12 Physical	SF-12 Mental
Warmth subscale	−0.003	0.186	−0.028	0.412 **	0.181	0.239
Conflict/criticism subscale	0.035	0.195	−0.098	0.292 **	0.415 **	0.402 **
Total	0.018	0.195	−0.073	0.356 **	0.318 *	0.359 *

* *p* < 0.05; ** *p* < 0.01; 6-CIT: 6-item Cognitive Impairment Test; ADAS-Cog: the Alzheimer’s Disease Assessment Scale–Cognitive subscale; INP: the Neuropsychiatric Inventory; QCPR: Quality of the Carer–Patient Relationship scale; QoL-AD: Quality of Life in Alzheimer’s Disease; SF-12 physical: the Short Form-12 Health Survey–physical dimension; SF-12 mental: the Short Form-12 Health Survey–mental dimension.

## Data Availability

The data presented in this study are available on request from the corresponding author.

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
