# Peer review of "The Quality of Carer–Patient Relationship Scale: Adaptation and Validation into Portuguese"

_ijerph, 2021, doi:10.3390/ijerph18031264_

Round 1

Reviewer 1 Report

Thank you for submitting your manuscript on the adaptation and validation of a quality of carer-patient relationship scale for publication consideration. The study addresses an important topic: that of psychometric validation and translation of a quality of carer-patient relationship scale. The scarcity of international research literature on scales of this nature and the importance of assessing quality of caregiver-patient relationships underscores the need for this study. The methods used to validate the measurement scales are appropriately described and the data is clearly presented. I just had a few, minor questions:

What are the qualifications of the 2 member panel referred to on line 75 that developed the preliminary version of the scale in Portugese?

How many people were on the expert review committee that evaluted the content validity of the QCPR version B and included 8 nurses (lines 83 to 87).

How was the benchmark of 0.8 determined for the index of content validity?

Approximatley 6 of the 25 references are recent (less than 5 years). Was this because of the paucity of recent literature on this topic?

Author Response

We would like to thank the reviewer for his/her appreciation and contributions that made this article clearer.

Reviewer - What are the qualifications of the 2 member panel referred to on line 75 that developed the preliminary version of the scale in Portuguese?

Answer: The required information was added. Please, see line 86.

Reviewer - How many people were on the expert review committee that evaluated the content validity of the QCPR version B and included 8 nurses (lines 83 to 87).

Answer: We think we don't explain the idea well. The sentence was corrected. Please, analyze lines 92 to 96.

Reviewer - How was the benchmark of 0.8 determined for the index of content validity?

Answer: The choice of the benchmark of 0.8 followed a theoretical framework that addresses this issue. The corresponding reference was included in the article.

Reviewer - Approximately 6 of the 25 references are recent (less than 5 years). Was this because of the paucity of recent literature on this topic?

Answer: Yes, very well observed. We were quite surprised by the absence of recent publications on this construct. We consider that this topic has been little studied.

Thank you for spending your time.

Reviewer 2 Report

First of all, I would like to congratulate the authors for choosing such an interesting topic and how little society in general takes it into account.
It is very important that our elders are well cared for and this is only possible thanks to the caregivers. Therefore, the caregiver-patient relationship must be good and therefore it is necessary to mediate the quality of this relationship correctly.

The methodology developed is correct, the results are clearly stated, and the discussion and conclusions are perfectly based on the results.

To improve, I would suggest expanding the introduction section a bit more.
It would explain how other countries have validated this same scale for other languages ​​and how it has worked.

I think a good job has been done

congratulations.

Regards

Author Response

We would like to thank the reviewer for his/her appreciation and contributions that made this article clearer.

Reviewer - To improve, I would suggest expanding the introduction section a bit more.
It would explain how other countries have validated this same scale for other languages ​​and how it has worked.

Answer: The information was added; please, see lines 50 to 57.

Thank you for spending your time.
